# Simulation and Sensitivity Analysis for Various Geometries and Optimization of Solid Oxide Fuel Cells: A Review

**Mina Tonekabonimoghaddam** [1] **and Ahmad Shamiri** [2,*]

1 Chemical Engineering Department, Faculty of Engineering, University of Malaya, Kuala Lumpur 50603, Malaysia; mina.tmoghadam@gmail.com
2 Reliability and Engineering Department, Dyno Nobel Moranbah, 667 Goonyella Road, Moranbah, QLD 4744, Australia
* Correspondence: ahmad.shamiri@incitecpivot.com.au or ahmadshamiri@gmail.com; Tel.: +61-7-4840-3708

**Abstract:** Solid oxide fuel cells (SOFCs) are considered as one of the most promising fuel cell types for application as high efficiency power generators. This work reviews the use of computational fluid dynamics (CFD) to maximise SOFC performance and life, and minimise cost, by considering numerous configurations and designs. A critical analysis of available literature proves that detailed research on the simulation of thermal stress and its damaging impact on the SOFC is still in its early stage of development. Numerical simulation is expected to help optimize the design, operating parameters and fuel cell materials. Therefore, sensitivity analysis of fuel cell parameters using simulation models is analysed to address the issue. Finally, the present status of the SOFC optimization efforts is summarized so that unresolved problems can be identified and solved.

**Keywords:** solid oxide fuel cell; computational fluid dynamics; sensitivity analysis; optimization





## 1. Introduction

State-of-the-art solid oxide fuel cell (SOFC) systems are designed for both small and large-scale stationary power generation systems [1]. This is because the SOFC is simple with high energy efficiency, can work with different hydrocarbon fuels, and can at least partially reform hydrocarbon fuels internally [2–6]. They are generally considered ideal for stationary applications with a high power range, i.e., for several 100 kW to the MW region, such as systems from LG Fuel Cell Systems or Bloom Energy [7], or for power outputs of 1–20 kW, such as the systems from HEXIS, Ceramic Fuel Cells Limited, or Versa Power [8]. Due to the higher power density of SOFC systems compared to other fuel cell types, SOFC systems have also been proposed for portable applications with power ranges of about 20–250 W [8]. Specifically, Delphi Automotive Systems is developing an SOFC that will power auxiliary units in automobiles and tractor-trailers [8,9]. Similarly, Topsoe Fuel cell manufacture SOFCs for stationary combined heat and power (CHP) in steps of 6 kW for residential applications [9,10].

SOFCs have the ability to produce all the required electricity to allow the engine to be smaller and more sufficient due to a high operating temperature. The SOFC can run on the same gasoline or diesel engine and maintains the running of all necessary electrical systems while the engine shuts off when not needed. Rolls-Royce Fuel Cell Systems Ltd. is developing a SOFC gas turbine hybrid system fuelled with natural gas for power generation applications in the range of a megawatt (e.g., Futuregen) [10,11]. Acumentrics, a US company, manufactures SOFC power generators for off-grid applications at a power range of 250 to 1500 W [11,12].

The application of an SOFC is bounded by some limitations, however, which are primarily due to its elevated operating temperature (500–1000 °C), relatively expensive construction materials (in some cases), and sensitivity to thermal stresses within the cells. Major research efforts have been placed on the materials, cell and stack designs, and

improvements in power density along with lowering the operating temperature down to 500–700 °C [13–15]. Most SOFCs are designed to function at elevated temperatures, thereby generating higher cell efficiencies in comparison to other kinds of fuel cells when operating on hydrocarbon fuels [16]. However, many research groups are working to reduce the operating temperature of SOFCs to under 650 °C in order to reduce material degradation, prolong stack-lifetime and decrease stack material costs by enabling the use of common metallic materials [17–19]. Therefore, it is of paramount importance to manage the stack temperature of the SOFC in a uniform manner, particularly for planar SOFCs [20]. To enhance SOFC technology for widespread market penetration, the system should demonstrate extended cell lifetime at reduced cost.

Modelling, simulation, control and optimization of SOFCs have become an important tool to understand, predict and enhance performance, and a means for improving lifetime through dynamic analysis of critical variables and advanced optimization algorithms (Figure 1) [21]. As shown in Figure 1, computational simulations are necessary to have a direct "insight" into the system and to allow the analyses of micro-structure, residual stains and electrochemical effects on macro-structural design, thermo cycling, transport phenomena, different component designs, stack and cell configurations and layouts, fuels, as well as to determine optimum operating conditions. This reduces the number of experimental tests to be performed, and costly and time-consuming physical prototyping methods [21,22]. Moreover, owing to the dynamic model, it is possible to analyse both the time response of fuel cell micro/macrostructure and its full function, thereby allowing the optimization of the system and its control logic. Understanding the fundamental mechanisms is an important step towards SOFC design optimization and performance improvement. In this respect, computational simulations have proved to be a cost-effective method. Many simulation investigations in the literature have been reviewed in this work in order to deliver performance prediction and parameter optimization. Some papers have studied the models of SOFCs for performance optimization [23–30], including the steady- and dynamic-states. Ramadhani et al. [26] have discussed the application of SOFC, in particular, the optimization strategies. They reviewed the decision variable, objective analysis, constraint, method and tools. However, the study did not focus on the sensitivity analysis. A number of recent qualified and comprehensive research and review papers were published with emphasis on different topics of the SOFC which are discussed in the next sections. The rapidly growing number of citations related primary research articles related to optimization of SOFC (Figure 2) is an indication that more comprehensive reviews in this field are necessary in order to draw general conclusions and provide some guided perspectives for future research. As indicated by the number of citations per year, interest is still growing (see Figure 2).

Neither of the previous studies focuses on the comprehensive sensitivity analysis for various geometries of solid oxide fuel cells. Since there has been no extensive review on this subject, the aim of this work is to provide a comprehensive literature review on the simulation, sensitivity analysis for various geometries and optimization of solid oxide fuel cells.

A sensitivity analysis of fuel cell parameters has been performed to capture the variables that most affect system cost and efficiency. Different simulation methods focussing on fuel cell behaviour prediction are discussed and reviewed herein. This work also summarizes the present status of various SOFC optimization efforts so that unresolved problems can be identified and settled.

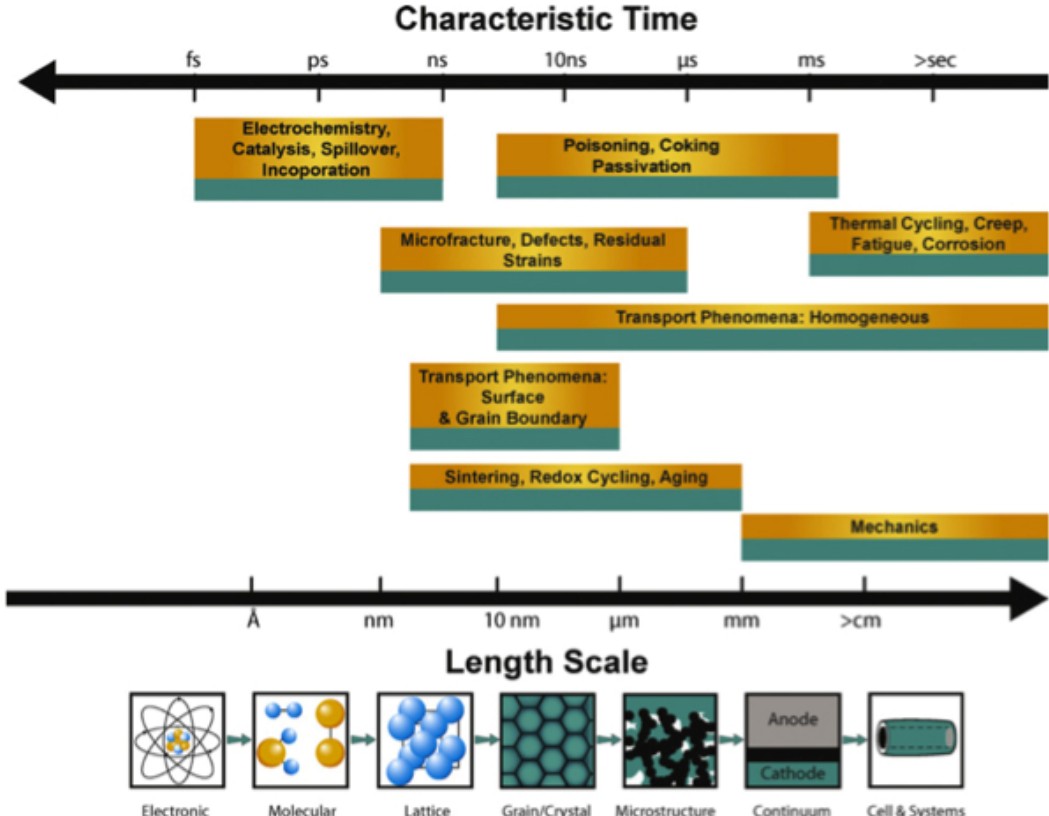

**Figure 1.** Different processes and their corresponding length scales and time frames during the operation of an SOFC (mathematical modelling takes all such processes into account) (figure reprinted from Grew et al. [21] Copyright (2012), with permission from Elsevier).

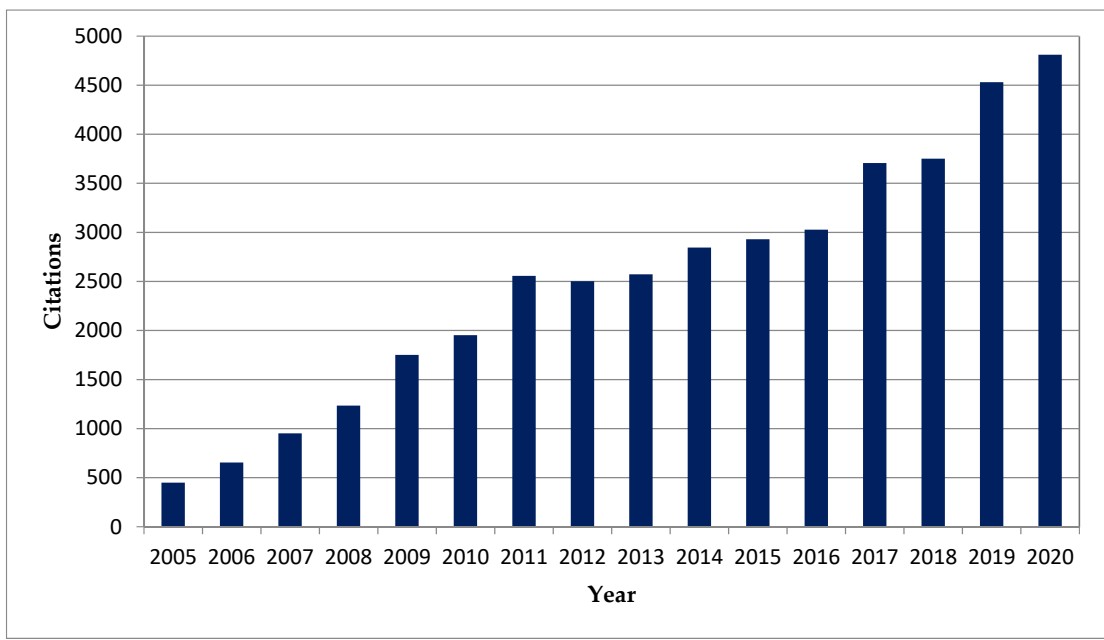

**Figure 2.** Number of citations of articles per year (until December 2020) related to "SOLID OXIDE FUEL CELL OPTIMIZA-TION", showing the increasing research interest in this topic. Data from Scopus.

## 2. SOFC Configuration Design

The first step of SOFC simulation is geometric design for creating a computerized model of the system that accurately represents its physical dimensions. Furthermore, in

the last decade, there has been numerous research endeavours aimed at increasing the performance of single SOFC systems through configuration design [24,25,31–38].

There are two main classifications for single SOFCs, which are: (i) self-supported, and (ii) externally-supported [31]. As for the self-supported category, the layer that is the thickest functions as support for the structural cell; hence, it could be adopted as electrolyte-supported, anode-supported or cathode-supported (as illustrated in Figure 3). On the other hand, in that of externally-supported, thin layers of single cell are set upon substrate that is porous in nature. In fact, these configurations possess both benefits and drawbacks. For example, configuration based on electrolyte support has a rather solid structure, hence, suggesting more resistance towards oxidation of anodes or reduced cathodes. Nonetheless, due to increased resistance from the electrolyte sheet that is quite thick, the temperature demanded is also high in order to lessen losses of ohmic in electrolyte [32]. In addition, when thin electrolyte is used, the temperatures used may not be high, especially when employing support from anode or cathode. However, it is challenging to develop layers of electrolyte that are dense and those that permeate gas, primarily when the electrolytes become thinner. Compared with the cathode-supported SOFC, there is more attention being given to the anode-supported configuration. Despite that, the low cost of cathode supporting materials may outweigh its disadvantages. Furthermore, when fuel cells are operating with hydrocarbon fuels, a low steam-to-carbon ratio in a relatively thin anode layer prevents carbon deposition on catalyst surfaces. A summary of the characteristics of different self-supported SOFCs is given in Table 1. The simulation studies of SOFCs based on self-supported designs are discussed in detail in Section 4.

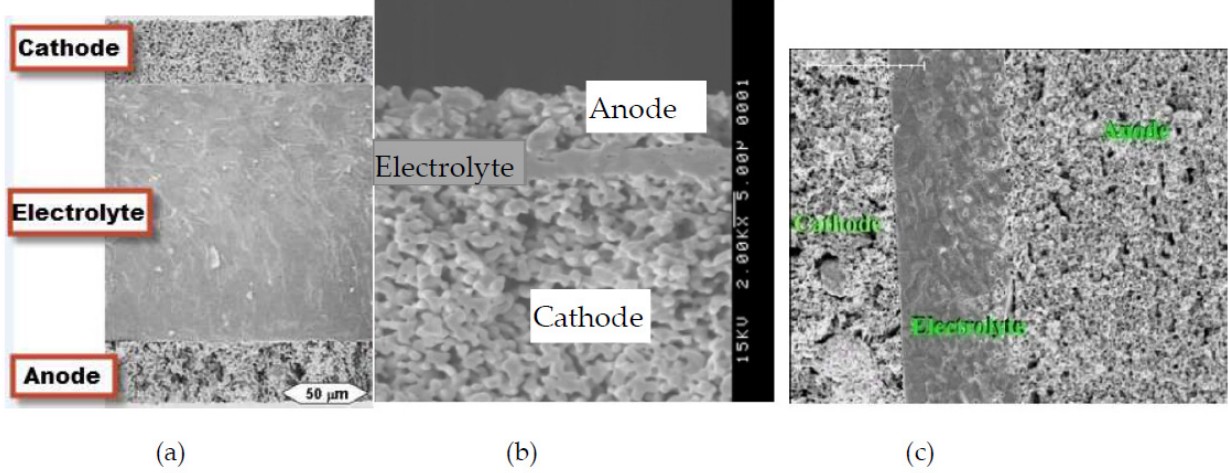

|      (a)      |      (b)      |      (c)      |

**Figure 3.** SEM micrograph of a single cell (electrolyte: YSZ in center, cathode: porous LSM, anode: porous NiO-YSZ): (**a**) electrolyte-supported, (**b**) cathode-supported, (**c**) anode-supported.

**Table 1.** A summary of the characteristics of different self-supported SOFCs.

| Cell Configurations | Cell Characteristic | Thermo Mechanical Issues: Thermal Expansion Mismatch Stresses and Propensity to Cracking or Delamination |
| --- | --- | --- |
| Electrolyte-supported | High Ohmic Contribution Low Cathodic Concentration Polarization Low Anodic Concentration Polarization | Minimal Tendency for Delamination due to Thermal Expansion Mismatch |
| Cathode-supported | Low ohmic contribution High cathodic concentration polarization Low anodic concentration polarization | Minimal Tendency for Cracking or Delamination due to Thermal Expansion Mismatch |
| Anode-supported | Low ohmic contribution Moderate anodic concentration polarization Low cathodic concentration polarization | Potential for Electrolyte Film Delamination exists as the YSZ Film is in Biaxial Compression |

## 2.1. SOFC Design

There are two principal SOFC configurations: planar and tubular (as shown in Figure 4). In a planar SOFC, the main components are an anode, electrolyte, a cathode and an interconnection between them. The components are all laminated in a plate-type structure. Most fuel cell manufacturers concern themselves with the planar SOFC configuration due to their advantages of low-cost, simplicity, high volume manufacturing as well as high volumetric power densities [33]. Within the electrolyte-supported planar SOFC, the electrolyte generally is made of YSZ with thicknesses of about 100 to 250 μm and an area of $10 \times 10$ cm$^2$ or larger as the supporting part of the cell. The operating temperature of this type of configuration is generally between 850–1000 °C due to the relatively high ohmic resistance of the thick electrolyte. Sealing is a significant factor for a planar SOFC. It inhibits gaseous leakage and reduces cell damage due to the thermal expansion of the cell stack during operation. As a consequence of the cell stack design, which needs a cell support to sustain the air tautness between the air and fuel flow channels, the application of the sealing process tends to be difficult.

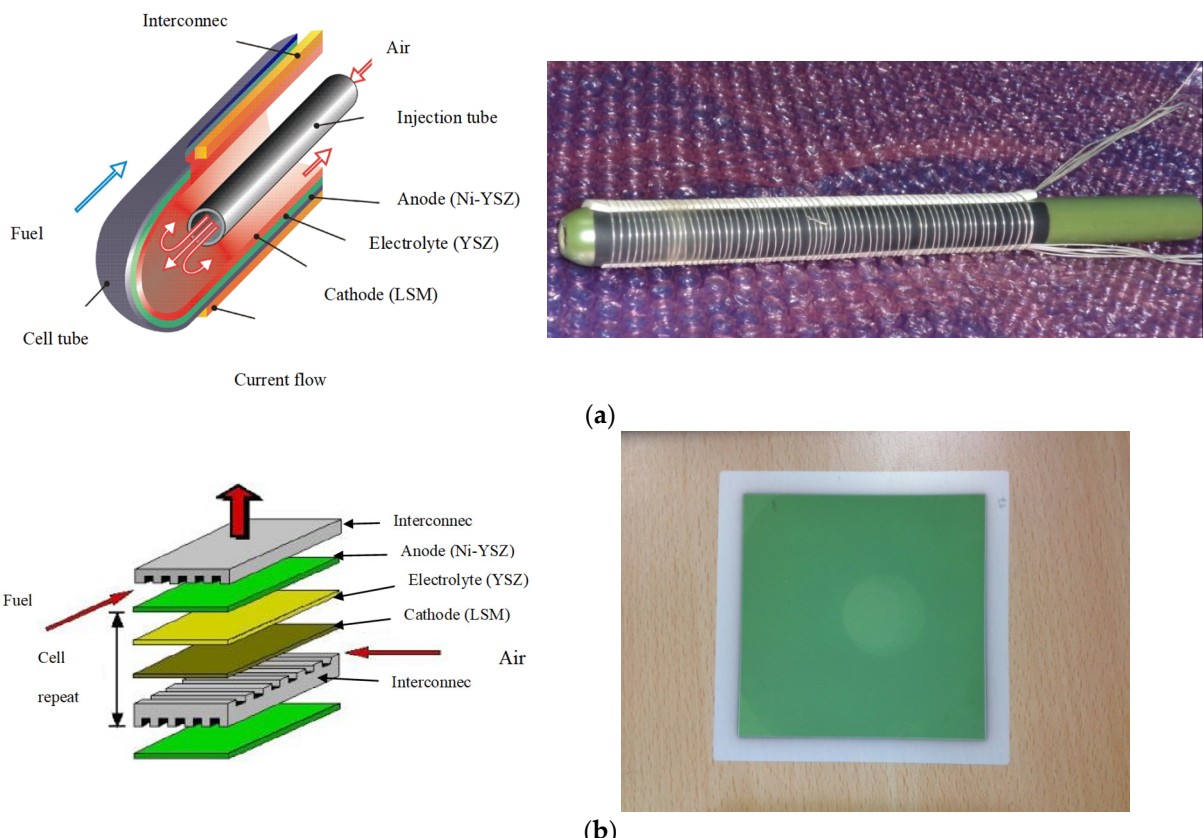

**Figure 4.** Schematic of: (**a**) tubular SOFC and (**b**) planar SOFC (figure reprinted from Hajimolana et al. [6] Copyright (2011), with permission from Elsevier).

Planar SOFCs are usually favoured by companies such as Sulzer Hexis. Meanwhile, as for the design of Sulzer Hexis, the supply of fuel is concentrated at the middle of circular cells with 120 mm diameter and supported by electrolyte. After that, the fuel moves towards the cell outer rim to burn, especially fuel that has yet to react with cells. Precisely, the air is supplied, gets heated, and then, moves to the middle part.

The tubular SOFC consists of the same components as a planar SOFC, but with a different structure. A major advantage of the tubular SOFC is that it eliminates the sealing problem, which improves thermal stability, in comparison to the planar configuration [35]. The sealants are required to satisfy all the yardsticks for all the components. They must be

durable in a variety of oxygen partial pressures (air and fuel) while diminishing thermal tensions during high-temperature performance. The trait of the seals should be excellent, since even tiny leaks in such seals may affect the cell potential, causing a decrease in operation. Sealant development is additionally complicated because the optimal sealant depends on the materials of other components resulting in stresses that are induced by a thermal expansion mismatch. Furthermore, this drawback leads to a slow start up capability and less thermal cycling sustainability [34,35].

A number of firms like Westinghouse/Siemens and US-based Acumentrics prefer tubular SOFC to be operated in both manufacturing and army arenas [11,12] for the ease of piling the single cell; hence, this requires low cost for processing [34]. In fact, conventional operations are run on normal fuels, for instance, propane and natural gas, while army related operations employ special fuels like those used for liquid logistics [11,12]. In another instance, a design with standard tubular was employed by TOTO, a firm based in Japan, which was actually initiated with lower industrial cost [35,36], in which 0.5 m length and 16 mm diameter short tubes were selected. Nonetheless, the Japanese firm, Mitsubishi Heavy Industries, has pursued another type of tubular design [6,35,36,38,39]. Besides, the cells that are single are placed in the middle part of the porous support tube, where a series electrical connection is given through interconnector rings made of ceramic, thus increasing the voltage of the tube. In this case, the fuel is pumped into the tube, with air on the external. Moreover, the findings obtained by Singhal [34], Williams et al. [40] and Godfrey et al. [41] portray lower current density in the SOFC tubular, in comparison to that of planar. This occurs due to the in-plane path, where the electrons move from electrodes circumference to interconnecting cells. This in turn increases internal ohmic losses.

Due to the need to increase efficiencies of chemical to electrical energy conversions with less impact upon the environment, several innovations in SOFC technology have been made [34,42]. To date, studies concerning power density escalation have generated a design that purports a flat tube with high power density SOFC (HPD-SOFC), especially to increase power density, but at the same time, maintaining its seal of security [34]. In another case, the Kyocera Corporation developed a design that is supported by anode [41–43]. Moreover, in order to hinder loss of current in the conventional designs of SOFC, the HPD-SOFC has been looked into, in which the design of tubular geometric is redesigned by giving it a flat look and attaching some elements to the electrode that gives flow of current [13]. A flat-tube SOFC has the same components and working principles as that of a tubular SOFC. It is comprised of a cathode, an anode, an electrolyte and interconnections. The design of the geometry and the structure of the stack are the differences between a tubular SOFC and a flat tube SOFC. The flat tube design has a seal-less feature compared to the tubular SOFC. Moreover, a flat tube SOFC can be configured with a thin cathode layer to decrease the concentration polarization, because the internal ohmic resistance can be reduced by adding ribs into the cell stack [13]. In fact, one advantage refers to the limited space that is void among the stacks of cells mainly because of the geometry, hence, making the bundle of cells tighter. As such, the scheme with a tube that is flat should project enhanced progression when compared to the SOFC, although the fuel and air supplies reflect similar approaches. This attribute decreases the resistance of the cell, but heightens the density of cell power.

In another innovation, a joint development between Mitsubishi Heavy Industries and Chubu Electric Power Company manufactured a combination of tubular and planar designs into a so called MOLB-type (Mono-block layer Built) planar SOFC [44]. The cells were manufactured up to a size of $200 \times 200 \text{ mm}^2$. This may not only increase the effective active area but also make the cell design fairly compact. Figure 5 shows the schematic of a monolith and an HPD SOFC.

Rolls-Royce (now LG) has developed an integrated planar SOFC design using a multi-cell MEA concept (Figure 6) [45], which takes advantage of both excellent thermal expansion compliances from tubular design and low-cost component fabrication from the planar configuration. For instance, a new symmetrically bi-electrode supported cell (BSC) was designed by Cable and Sophie by employing an approach known as the freeze tape

casting method [46]. This fresh BSC concept possesses dual exceptional attributes, which are as follows: (i) more stable properties of the mechanical are ensured in the conditions for thermal cycling, and (ii) thick and porous electrodes substitute the flow channels of gas and fuel in order to provide a more compact cell layer. Nonetheless, such tight layer poses an issue, which affects the efficiency of the gas and fuel supply, mainly due to the resistance exerted by the electrodes that are porous.

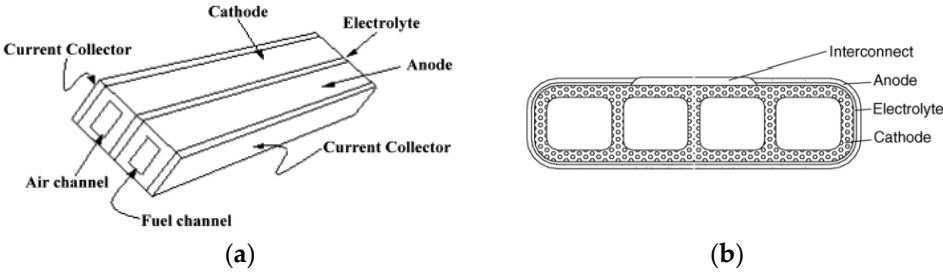

(a)  (b)

**Figure 5.** Schematic of the: (**a**) monolith type fuel cell modelled and (**b**) HPD-SOFC (figure reprinted from Singhal, S.C [34] Copyright (2000), with permission from Elsevier).

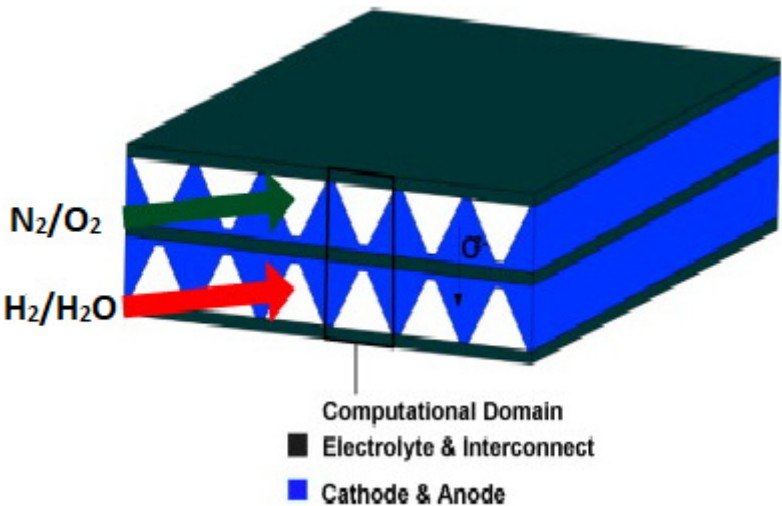

**Figure 6.** Multi-cell MEA concept developed at Rolls-Royce (figure reprinted from Gardner et al. [47] Copyright (2000), with permission from Elsevier).

Meanwhile, a fresh SOFC was looked into by Shi and Xue [13] by applying a simulation with a numerical basis, whereby BSC electrodes that are porous and thick are attached to channels that are micro in size, as illustrated in Figure 7. Thus, it was projected that minute-sized channels would improve the supply of gas and fuel by dismissing the ridiculous pressures of gases, besides taking advantage of the compact and symmetry features of the BSC.

The fuel cell developed by Ceres Power is a unique adaptation of the SOFC technology, which uses a new generation of metal supported ceramic cells. This type of fuel cell invented by the Imperial team is particularly well suited to operation with hydrocarbon fuels such as natural gas [45,46,48]. These properties mean that the fuel cells can run at substantially lower temperatures than conventional designs. Lower temperatures mean that an extremely thin layer of stainless steel can be used in the fuel cells, dramatically reducing material costs, improving resilience and increasing power density, thus making them more attractive for domestic applications such as micro combined heat and power (mCHP).

A very new innovation made at Imperial College London aims to address outstanding technological and economic issues by establishing the feasibility of, and developing, a novel design of SOFCs [46,47,49]. Moreover, fibres that are hollow are used to generate this SOFC, hence, adding to the surface area among the electrodes, which later escalates

power output per unit volume/mass and ensures higher temperatures sealing; and most importantly, these fibres decrease the costs incurred (Figure 8).

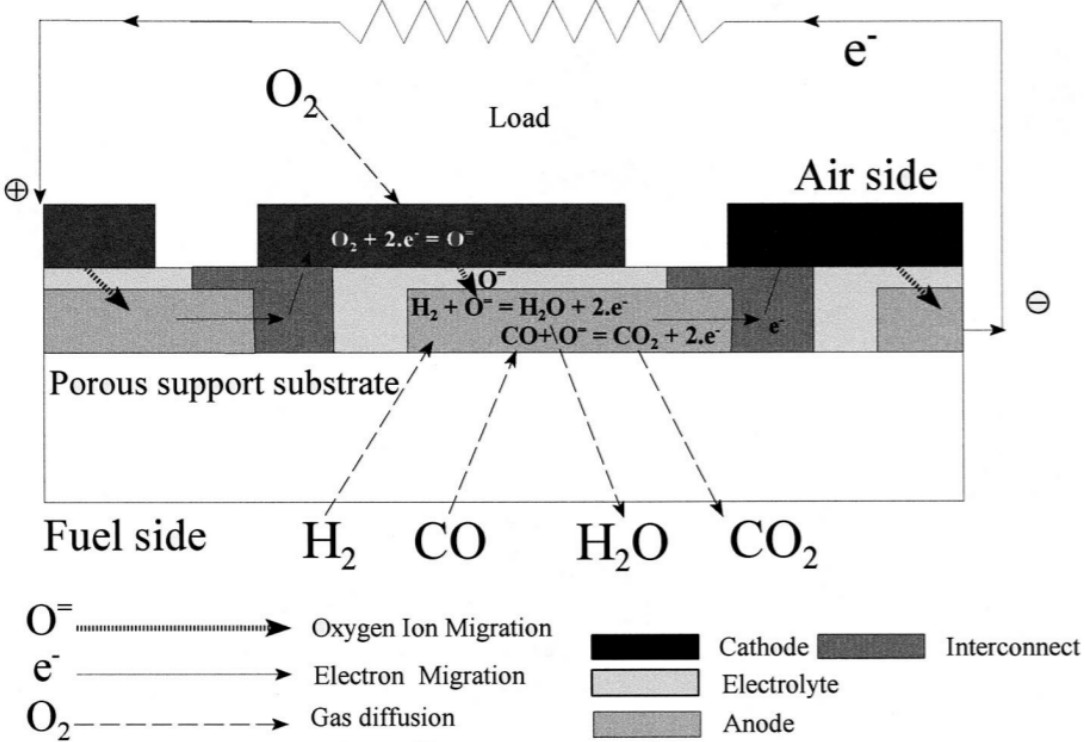

**Figure 7.** Schematic of BSC with micro-channels (figure reprinted from Gardner et al. [47] Copyright (2000), with permission from Elsevier).

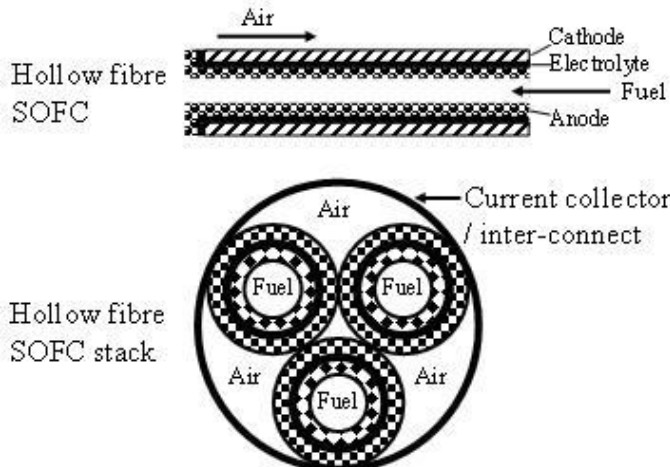

**Figure 8.** Hollow fiber SOFCs designed at Imperial College London (figure reprinted from Gardner et al. [47] Copyright (2000), with permission from Elsevier) [46,47,49].

### 2.2. Micro SOFC

A new concept of micro-SOFC (μ-SOFC) has been looked into, for several reasons: high SOFC power densities, as well as advancement in film and micro technologies [50–53] (Figure 9). In fact, this μ-SOFC requires low power within 1–20 W, similar to the use of a battery in low energy electronic devices like camcorders, industrial scanners, medical devices, battery charger, and laptops. Additionally, this μ-SOFC design offers energy density up to four folds and optimum energy/weight, when compared to batteries that could be charged like Li-ion and Ni metal hydride batteries [54].

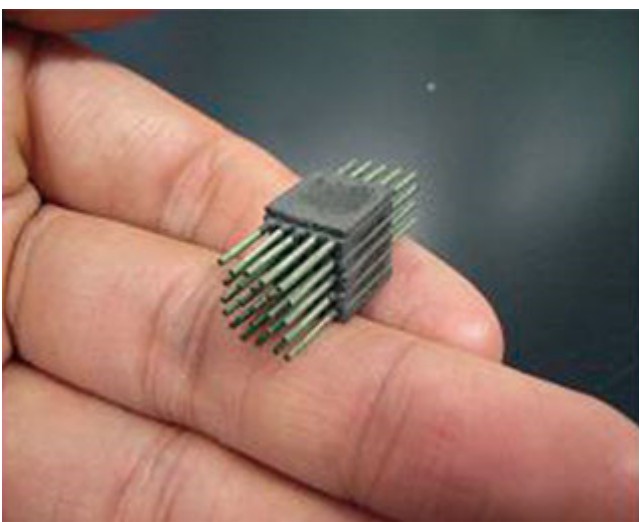

**Figure 9.** Schematic of a micro-tubular SOFC (figure reprinted from Hanna et al. [55] Copyright (2014), with permission from Elsevier).

The thinner SOFC layers reduce the ohmic loss at lower operating temperatures. In addition, thinner layers reduce the thermal mass of the cell [56]. Therefore, a stack of μ-tubular SOFCs is suitable for rapid start-up and shut-down cycles, with only a few minutes compared to hours for conventional designs [57]. μ-tubular SOFCs have been heated from room temperature to 850 °C within 10 min without failure and survived thermal cycling [58].

The small diameter of the μ-tubular SOFCs allows for the 'packing' of a more 'active area' within the same volume, thereby increasing the volumetric power density of a stack. The latter two advantages are especially important for portable power generation applications [57].

Nevertheless, this μ-SOFC projects decreased cell functionality because of the low flow of gas within the micro tube [59], which leads to deprivation of fuel, hence generating a decrease in electromotive force, as well as escalated resistance against anodic polarization. Therefore, some material with porous attribute within the free space could enhance the flow of gas to the cells [59].

Thermal management is a critical issue in the development of μ-SOFCs. In order to control the temperature of a SOFC during single cell testing, a furnace was constructed by Kendall and co-workers [60]. The furnace comprised two refractory bricks that were hollowed out to house electrical elements and a small manifold to hold a micro-tubular SOFC. The heating elements were constructed from Kanthal ribbon, with a resistivity 5.476 $\Omega\,\mathrm{m}^{-1}$, surrounding the supporting alumina tubing. These heating elements were connected to a power supply fitted with a temperature control device. Simulation approaches can help in the specification of operating conditions and designs that minimize temperature gradients and maximize the use of heat produced in the SOFC stacks. However, few investigations address the issue of thermal stress in μ-SOFCs [58,61–63]. Amongst them is the study conducted by Lockett et al. [58], who investigated the thermal management of a 20-cell μ-SOFC stack using a CFD model.

Different simulation methods have been studied for μ-SOFCs with different designs and operating conditions to enhance their performances and energy efficiencies. Aimed to improve the current efficiency, a new design of a μ-tubular SOFC was proposed [64]. In addition, some hollow fibres with ceramic are used in μ-SOFC to function as supportive element, where the functional layers are laid above. Furthermore, the conditions of thermal-fluid and electrochemical proposed in the design are examined via simulation of CFD, which revealed enhanced performance mainly due to essential decrease in the losses of ohmic, in comparison to micro tubular design supported with anode. In a different study, a parametric analysis was performed to study the effects of temperature, fuel flow rate, fuel composition,

as well as anodic and cathodic pressures on the μ-tubular SOFC performance [65]. Besides, this model of CFD incorporates several features like mass, momentum, species and energy balances, as well as transfer of ionic and electronic charges. In fact, this model was applied in a market-available multi-physics software programme known as COMSOL 3.4 by employing the method of finite element in order to determine the equations of partial differential. Moreover, it was projected that for output current of 0.53 A cm$^{-2}$ at 550 °C; an ionic current density at 0.65 A cm$^{-2}$ would have to be generated by the fuel, in which the variance had been due to leaks that occurred at the internal current. Additionally, temperatures below 500 °C did not project any significant leakage of electronic current.

A CFD simulation based on a multi-dimensional approach, coupling heat transfer, fluid-dynamics and electrochemistry of increasingly complex systems (micro and midi-reactors, constituting 15 and 45 tube assemblies, respectively) was performed to design a μ-tubular SOFC [66]. The system was designed to optimize the geometry of the respective modular assemblies in terms of air and heat release management to acquire the maximum performance in terms of power density whilst keeping the impact of thermal stresses to as low as possible with regards to cell durability. In another effort, a 3D thermal-fluid-dynamic model of a micro-tubular SOFC was presented to analyse both micro- (i.e., a 15 tube assembly) and midi-assemblies (up to 45 tubes) with the aim of optimizing the fuel cell configuration [67]. A summary of the characteristics of different SOFC designs is given in Table 2. Although several innovations to the SOFC in terms of design and configuration have been made to increase the power and performance, still more attempt is required to achieve this goal.

**Table 2.** Summary of different SOFC design characteristics.

| SOFC Design | Characteristics Design | Advantages | Drawbacks | References |
|---|---|---|---|---|
| Planar SOFC | The components are all laminated in a plate-type structure | Low-cost, simplicity in manufacturing and high volume manufacturing with high volumetric power densities. Has higher energy efficiency compared to tubular SOFCs | Difficulties in sealing Internal stresses in cell components due to non-uniform temperature distributions and high manufacturing cost | [33] |
| Tubular SOFC | The components are all laminated in a tube-type structure | Easy to assemble into a cell stack, leading to a lower processing cost | Has a much lower current density than a planar SOFC High ohmic loss | [35] |
| Flat-tube | The tubular geometric design has evolved into a flattened one with ribs in the electrode acting as bridges for current flow | Enhanced power density compared to planar and tubular SOFCs Has a seal-less feature compared to the tubular SOFC Can be configured with a thin cathode layer to decrease the concentration polarization The void space between the cell stacks is reduced | Retains the feature of secure sealing | [34] |
| Bi-electrode supported cell | Uses a freeze tape casting technique | More stable mechanical properties in thermal cycling conditions | This compact design imposes a great challenge on how to effectively feed fuel/gas due to the high diffusion resistance of porous electrodes | [46] |
| μ-SOFCs | | Higher energy densities per volume and specific energy per weight are obtained compared to rechargeable batteries A stack of μ-tubular SOFCs is suitable for rapid start-up and shut-down cycles, with only few minutes compared to hours for conventional designs Low cost Low concentration | Lower cell performance | [50–53] |

## 3. Computational Simulations of SOFC

Hence, comprehending the basic functions is deemed as an essential phase in both optimizing and improving SOFC and its functionality. Although many studies have looked into these SOFC designs [68–70], it is indeed challenging to carry out empirical studies, especially in determining the scenarios of internal transport within the intricacy of internal physical mechanisms. On top of that, SOFC related studies take up a lot of money and time, thus highlighting the importance of computational simulation that saves both cost and time [71].

In addition, optimum designs must be built to ascertain excellent functionality and consistency, as well as lowness in cost. Besides, good simulation models offer ideas about functionality, besides identifying the tools required in devising a design that is functional, which takes into account estimates accuracy, detailed information and varied simulations via computer, especially when building numerical models.

Numerical modelling and simulation must be used very carefully, however, especially when used for prediction purposes [72–75]. The models are typically based on computational and simulation approaches such as FLUENT, CFD-RC, gProm, COMSOL, MATLAB and Star-CD, to evaluate the cell performance at different conditions [71,76–80]. The simulation and computational models are coupled to the 0, 1, 2 and 3 dimensions.

Eichhorn Colombo et al. [81] employed a detailed solid oxide fuel cell (SOFC) model for micro-grid applications to analyze the effect of failure and degradation on system performance. They presented the design and operational constraints on a component- and system level. A degrees of freedom analysis identifies controlled and manipulated system variables which are important for control. They used experimental data to model complex degradation phenomena of the SOFC unit. They assumed that the SOFC unit is consisting of multiple stacks. They studied the failure scenario of the loss of one individual SOFC stack, e.g., due to breakage of sealing or a series of fuel cells. Their simulation results reveal that degradation leads to significant drifts from the design operating point. In addition, failure of individual stacks may bring the still operating power generation unit into a regime where further failures and accelerated degradation is more likely. It is presented that system design, dimensioning, operation and control are strongly linked. Apart from specific quantitative results perhaps the main practical contribution are the collected constraints and the degrees of freedom analysis.

Haoran et al. [82] developed a hybrid model for an on-line analysis of SOFCs at the cell level. Their model combines a multi-physics simulation (MPS) and deep learning, overcoming the complexity of MPS for a model-based control system, and reducing the cost of building a database (compared with the experiments) for the training of a deep neural network. They considered the maximum temperature gradient and heat generation as two target parameters for an efficient operation of SOFCs. Their modelling results reveal that a precise prediction can be achieved from a trained AI algorithm, in which the relative error between the MPS and AI models is less than 1%. Furthermore, an online optimisation is realised using a genetic algorithm, achieving the maximum power density within the limitations of the temperature gradient and operating conditions. Their method can also be used for prediction and optimisation of other non-liner, dynamic systems. Their study shows that the combination of an MPS, a deep neural network and a genetic algorithm provides a promising solution for model-based control systems for precisely and quickly analysing the performances of SOFCs and other non-linear systems. Their deep learning algorithm is trained based only on simulation data under stationary operating conditions. Further training with more experimental data on the algorithm is needed for a wider application under dynamic operating conditions.

Zero-dimensional (0D) methodology, also called box models, simply allows studying processes that can be analysed without taking into account spatial configuration and geometry. Furthermore, the system is investigated using a box and the spatial is averaged for the dimensions. Thus, any difference is dismissed, but the transfer scenario is weighed in to distinguish the output from input variables. Nonetheless, time emerges as the

independent variable for dynamic simulation. Hence, far from being intricate, this model is indeed applicable for SOFC design and does not require numeric. The dynamic variables, flow rates, gaseous compositions, pressures and the basic energy balance to determine the sources of heat to maintain the cell temperature are assumed to have changed between input and output values or in time (dynamic models) [62,83,84]. However, in these models the geometry does not affect the fuel cell performance directly due to the neglect of the spatial variation of variables. Therefore, 0D models can be used in studies where the focus is not on the fuel cell itself but on how the SOFC affects the performance of the entire system. 0D models are usually applied for thermodynamic modelling of SOFCs and numerical analysis of fuel cell-based power systems (such as SOFC/gas turbine systems and combined heat power systems) or µ-SOFCs (see Figure 10). The objective of this kind of analysis could be for optimizing the fuel cell performances and to determine the optimal operating parameters for the chosen application.

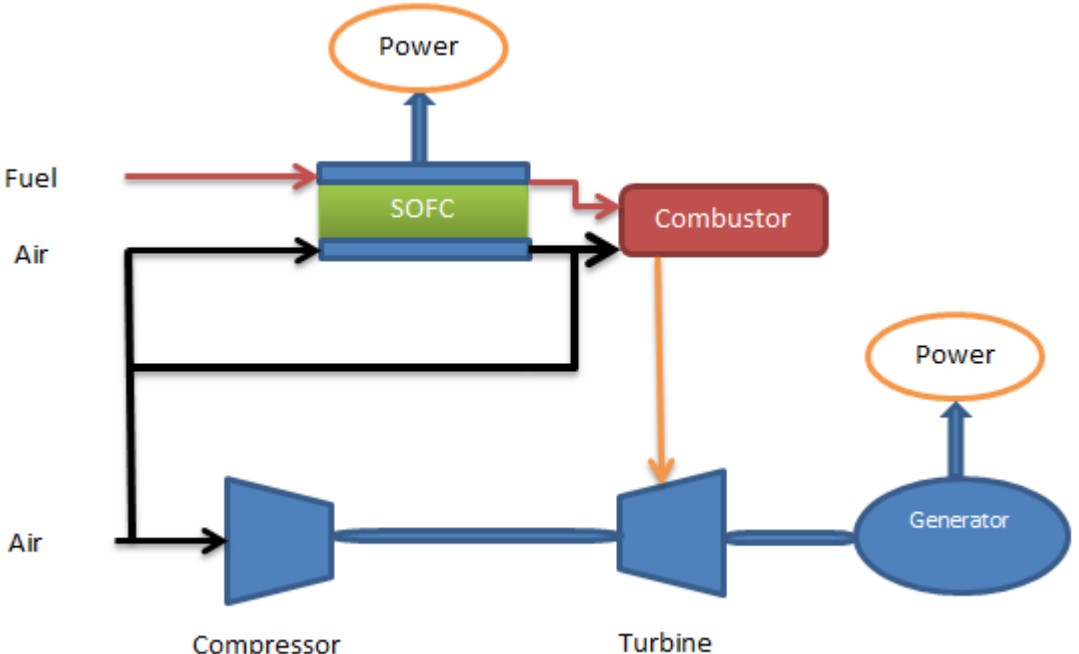

**Figure 10.** Schematic of a hybrid SOFC/GT system for 0D simulation models.

In parallel to the design and optimization of the SOFC/GT system, control and system integration solutions have also been explored in 0D models [85,86]. A simple control relevant model may perform better than its complex 3D counterpart for control design, which on the other hand, may be suitable for design and analysis of the process. A process expressed by a very complex model may be controlled by a regular PID controller. On the other hand, a 0D model may be attributed with plenty of environmental and economic constraints that require multivariate controllers to maintain the optimal performance of the system. Hence, a model-based predictive control strategy, which could handle various operational constraints, has been developed in a 0D model for coordinated power and thermal management [85,86].

In 1D simulation models, the fuel cell is represented by ordinary differential equations and the variation of parameters along the other two directions are negligible. In planar SOFCs, the dimension is usually determined by the gaseous flow direction in the fuel cell. In the 1D, the parameters vary along a direction parallel to the axes of the gas channels for planar SOFCs. Therefore, cross-flow planar SOFCs cannot be simulated by 1D simulation approaches [87].

For tubular SOFCs, the dimension is usually determined by the tube axis which includes the direction of the fuel and oxidant gas flow as in models presented by some

researchers [62,88,89]. Gaseous compositions, flow rates, pressures and temperatures are averaged for each gaseous flow channels. The electrochemical model is considered as a simple prototype.

A rather similar characteristic of current density distribution along the fuel cell is obtained for 0D and 1D models [84]. However, for lower current densities, an average temperature of the fuel cell attained with the 0D model is higher than that computed by means of the 1D model [84]. Besides, numerical findings highlight the significance of temperature for predictions of source and losses, as well as to model properties of materials and transport. Meanwhile, density of current is only required in estimating losses and source. Moreover, information about temperature reveals thermal stresses levels upon fuel cell, hence affecting both the design and the optimizing courses. Therefore, 0D models are not suitable for thermal stress studies.

In two dimensional approaches, two directions of the entire fuel cell operation are considered. In this method it is assumed that the fuel cell's behaviour is the same for each angular section. In case of three-dimensional approaches, mathematical formulae are usually written in the form of partial differential equations. A 3D model is suitable for sensitivity analyses of geometrical parameters of the fuel cell [90,91]. In fact, these researches are rather important in dictating the performance, besides identifying the associated drawbacks to be hindered. Nevertheless, a modelling that is in 3D is definitely costly for fuel cells with solid oxide in layers. Moreover, high power has to be used for implementation of 3D CFD models with layers of SOFC and support elements. Furthermore, time becomes a constraint, hence impeding optimization. Precisely, estimations that are accurate are sought when model optimization is carried out in relation to layers of SOFC, together with reformers or pre-heaters.

## 4. Sensitivity Analysis

In order to improve the efficiency and reduce the cost, sensitivity analysis of fuel cell parameters needs to capture the significant variables. The parameters that affect the efficiency and cost of fuel cells are: (i) fuel cell design parameters, (ii) operating parameters and (iii) fuel cell materials. Numerical simulation is expected to help optimize the design, operating parameters and fuel cell material. Therefore, sensitivity analysis for these parameters using simulation models is conducted to optimize the fuel cell systems operation with higher performance and lower cost to make them commercially viable in the market.

### 4.1. Sensitivity Analysis of Fuel Cell Design Parameters

Optimizing the fuel cell's configuration is a significant means towards enhancing its power density and operation. The optimization of an SOFC's external configuration includes an optimal fuel cell inventory allocation of cross-sectional area and the width of the gas channels so that the SOFC's electrical and pumping powers are balanced accordingly to attain a maximum net value. In addition, resolution of optimally active TPB regions in both electrodes as well as the electrolyte, which is founded on the trade-offs between all the potential losses in the electrodes is also significant for improving the power density.

Most SOFC designs have been reported using numerical modeling and simulation, with a goal to maximize the power density and fuel utilization [13,92–95]. Moreover, the issue of designing a reliable SOFC due to high temperature operation was investigated in details in the literature [2,34,35]. In addition, simulation helps to curtail the non-uniform current density and temperature distributions that directly influence the thermal stresses in a range of SOFC components [22,96–98].

Dang et al. [99] developed and solved a numerical model of gas flow, heat transfer, mass transfer and electrochemical reaction multi-physics field coupling of a planar SOFC. They analyzed distribution of velocity, temperature and concentration inside the SOFC cell. They discussed the influence of cathode inlet flow rate, porosity, rib width and other parameters on the performance of SOFC. Their simulation results showed that within a

certain range, increasing the cathode inlet flow rate can significantly increase the average current density of the cell. Increasing the porosity of the electrode can improve the gas diffusion of the porous electrode, thereby increasing the rate of the electrochemical reaction. Increasing the width of the ribs will result in a significant decrease in cell performance. Therefore, the rib width should be reduced as much as possible within the allowable range to optimize the working performance of the cell.

Parametric investigation of the SOFC flow channel arrangement is the focus of attention for optimal power generation. Significant parameters involve the widths of anodic and cathodic flow conduits, their heights and the width of current collector contacts with the cell. Table 3 summarizes the power density and electrical efficiency with various geometrical configurations of flow channels indicating with types A-F. Decreasing the channel sizes causes a shorter current path and also enables higher heat/mass transport rates [100]. This leads to variations in temperature and concentration distributions, and also affects the power efficiency of an SOFC. The conduits with smaller heights suffer a shorter current path and also result in significant improvement in the mass and heat transfer coefficients. Higher mass transport rates in smaller conduits reduce the concentration polarization in channel flows significantly and enhance the cell's operation. Generally, when the flow channel dimension is reduced from conventional to micro-scale, the temperature/concentration gradient of the SOFC drops, and the heat/mass transfer coefficient is observed to rise accordingly. In addition, minimized sizes are significant to developing low cost and portable SOFCs. Thus, it is expected that concentration loss and operating temperature in smaller SOFCs may be effectively decreased. However, for the smallest cell, the ohmic polarization is higher than those of the larger cells due to a lower operating temperature [101]. Therefore, an optimized fuel cell length should be designed.

**Table 3.** Power density for different geometric arrangements of the flow conduits at 0.6 V [98].

| Type | Configurations (mm) | | | | | | Power Density (mW/cm$^2$) | Electrical Efficiency |
| | Anode Channel Height | Cathode Channel Height | Anode Width | Cathode Width | Anode Rib Width | Cathode Rib Width | | |
|---|---|---|---|---|---|---|---|---|
| A | 1 | 1 | 1 | 1 | 1 | 1 | 752.2 | 49.2% |
| B | 0.5 | 1 | 1 | 1 | 1 | 1 | 758.5 (0.83%) | 49.6% |
| C | 1 | 0.5 | 1 | 1 | 1 | 1 | 754.9 (0.36%) | 49.6% |
| D | 1 | 1 | 2 | 1 | 1 | 1 | 755.5 (0.44%) | 49.3% |
| E | 1 | 1 | 1 | 2 | 1 | 1 | 753.1 (0.12%) | 49.7% |
| F | 1 | 1 | 1.4 | 1.4 | 0.6 | 0.6 | 767.4 (2%) | 50.7% |

Alteration in the size of annulus within electrode tubes at high current densities could give significant impact upon tubular SOFC, especially on polarization concentration and activation [102]. Thus, a drop in anodic channel radius for tubular SOFC supported with anode could escalate the decreasing channel pressure primarily due to heightened cell length, as well as a decrease in resistance of ohmic due to the current path that is shorter, but electrode thickness and electrolytes are retained [92]. Generally, difference in tubular SOFC radius brings variation to the electron path circumference length, while alteration in the length of cells modifies the cross-sectional zone for electron movement [103].

For the cell with the minimum rib resistance, SOFC performance enhances moderately with reduced channel dimension [103]. In contrast, cell operation enhances significantly with the decrease in channel dimensions for the case whereby the SOFC possesses higher rib resistances. Nevertheless, a cell that has the least dimension for channel disregards maximum output for the terminal. In fact, there may be two reasons for this occurrence: (i) lower temperature accelerates loss of ohmic at the solid area, hence associated to func-

tional temperature, and (ii) a current path that is shorter and a minute loss of concentration might partially or completely interact with progression of ohmic polarization [104].

A wider rib decreases the enhancement in the temperature rate. Therefore, it may be concluded that such an arrangement could reduce thermal stresses. Figure 11c,d show that for cells having thinner ribs, the oxygen concentrations are more constantly distributed across the whole porous layer (z-direction), which excludes inefficiencies that are usually caused by obstructed "dead zones" below the ribs [104]. When compared with a thicker rib, the oxygen molar fraction next to the surface of the diffusion layer is far higher for the cell with a thinner rib. The decrease in concentration loss for a thinner rib, however, is not significant enough to counteract the enhancement in ohmic loss and thus, the terminal output still reduces. Therefore, it is anticipated that a lower channel width ratio to the rib width may decrease the ohmic losses at the interface and enhance cell performance [104].

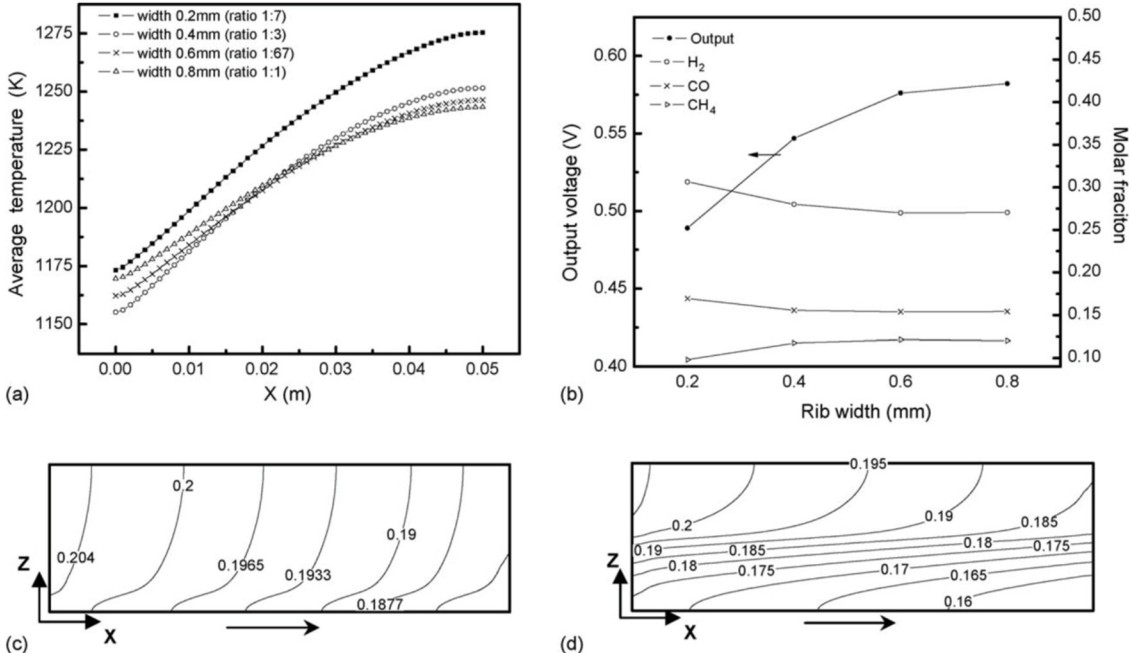

**Figure 11.** Effects of rib width on SOFC performance: (**a**) average top interconnector temperature, (**b**) comparison of performance, (**c**) oxygen at the cathode/electrolyte interface with the thinnest rib (ratio 1:7), (**d**) oxygen at cathode/electrolyte interface with the thickest rib (ratio 1:1) (figure reprinted from Ji et al. [104] Copyright (2006), with permission from Elsevier).

Rib number tends to improve the cell terminal voltage. This enhancement is higher when the rib number rises from a small to a median value. Further enhancement in the rib number, however, reduces the magnitude of the enrichment [105].

The ratio of the support thickness to the conduit diameter is taken as significant design parameter. A reduction in the anode support thickness results in an enhancement in the hydrogen mass transfer rate [105]. Simultaneously, the ohmic polarization tends to decrease as a consequence of the reduction in the electron path. The thin support may reduce the mechanical durability of the cell, however, leading to deformation or cleavage of the cell during manufacture, stacking or when running [105]. For an electrolyte-supported SOFC, a greater ohmic polarization in the electrolyte is a significant issue hence adopting a thin-film electrolyte to considerably decrease the ohmic loss is a valuable consideration.

Although there are many investigations on sensitivity analysis of fuel cell parameter design, still, more research studies are required on optimal design and geometry optimization of SOFCs to enhance the power density and fuel cell volume reduction. Studying novel fuel cell configuration design is necessary for fuel cell cost reduction and power density enhancement.

*4.2. Sensitivity Analysis of Flow Field*

For efficient cell stack design, simulation study of the uniform inlet flow is significant for reducing the temperature changes in the cell stack structure and enhances the cell power density [106–109]. Cross-flow geometries have minimal effect on gas flow mixing. They enhance cell active area, resulting in improved SOFC performance whilst maintaining thermal gradients and current path lengths to the standard planar fuel cell geometry [110].

Counter-flow configuration is the subject of interest as a consequence of its potential for higher efficiency. This is the case because of the SOFC's high efficiency in using the electrochemically generated heat for fuel processing. A design with counter-flow channel generates power density higher when compared to its related co-flow and cross-flow aspects [87]. Nonetheless, the design of layered cells employing channels with counter-flow portrays some integral inadequacies, including irregular flow, as well as intricate designs that demand fuel and air flow inlet and outlet coupling. Therefore, in order to decrease these drawbacks, an intricate configuration based on geometrics is deemed as significant for channels of inlet and outlet meant for fuel and oxidizer.

On top of that, the counter-flow has been discovered to be more efficient when compared to co-flow mainly due to the high current at the entrance area of the anode for its concentration of reactant [111]. In fact, for the same functional setting, higher voltage and bigger power output could be predicted for the design of counter-flow primarily because of low concentration and active polarization [112].

A good design of flow distributors can allocate the fuel and the air uniformly onto the anode and the cathode for achieving consistent diffusion processes through porous electrodes. It has been discovered that by simply employing small guide vanes equally spaced around the feed header of conventionally used rib-channel flow distributors (to effectively improve the degree of flow uniformity in flow distributors), the power density of the single-cell stack may be enhanced in comparison to that where guide vanes are employed under normal experimental conditions [100].

The enhancement of flow uniformity in the distributors is useful to enhance the redox stability of the Ni-based anode, so that a balanced employment of the anodic catalyst may be attained for the continuous cell operation. This is significant in extending the longevity of the cell stack [100]. Thus, the optimization of interconnects or flow distributors is crucial for further enhancement of the SOFC's performance.

*4.3. Sensitivity Analysis of Micro-Structures and Electrodes-Electrolyte Thicknesses for an Efficient Triple Phase Boundary (TPB)*

The essential aspects that accelerate the performance of cells are the provision of effective TPB or electrochemical reaction region, and increased conductivity of ionic and electronic in both ion and electron of the conducting elements [113].

The microstructural features considerably affect transport and reaction phenomena, influencing the electrode performance. For instance, decreasing the particle size substantially improves the number of reaction sites [114], which may be maximized by adapting the electrode composition [113]. The porosity is another significant factor: while a reduction in porosity initially causes an enhancement in electrode efficiency [115], blockage of pores and gas diffusion constrictions tend to hamper the electro-chemical performance to below a critical limit. Figure 12 clearly illustrates that undesirable microstructures such as insufficient porosity reduce cell performance [92].

Thickness is a geometric aspect that has an impact upon the performance of electrode [113] because an increment in thickness improves the amount of sites with reaction. Otherwise, electrons, as well as molecular and oxygen ions, would have to move further in order to arrive at/depart from the sites with reaction. Therefore, electrode thickness could affect both loss of ohmic and over potential concentration, thus influencing the performance of fuel cell [56]. Moreover, the anode thickness in a tubular SOFC could influence the surface area of TPB that involves reactions of both cathode and anode. Besides, an increment in electrolyte thickness alters the resistance of ohmic, which could even halt

the performance [116], as well as affecting the length of current path in cathode and the cathodic active reaction area [116].

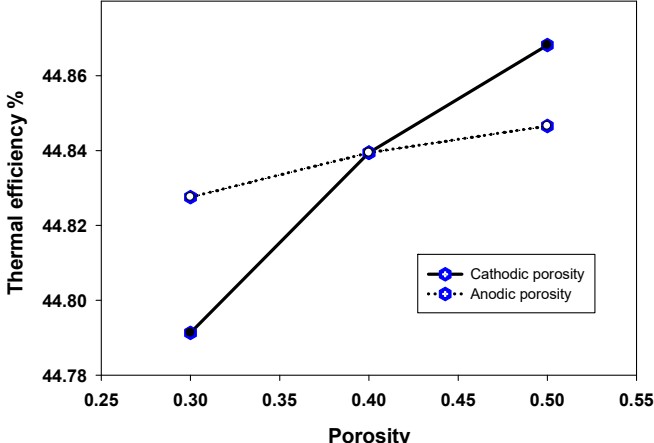

**Figure 12.** Effect of electrode porosity on fuel cell efficiency (figure reprinted from Hajimolana et al. [92] Copyright (2012), with permission from Elsevier).

In general, the over potentials of ohmic have to be hindered due to its significant effect upon the potential loss of cells [117–119]. Specifically, the ionic conductivity that derives from ion conducting elements at reaction region must be enhanced for cell functionality improvement to generate ohmic [120].

Meanwhile, due to charge transfer resistance and active polarization, over potential anodic becomes a factor, whereas the case of over potential of ohmic in anode takes place because of ions and electrons flow resistance at the reactive region and at the anode layer, respectively [62,83]. Besides, the potential of loss in cells often derives from the anode in the SOFC and the four times smaller anodic concentration over potential when at increased current density [120]. Other than that, the effect of cathodic concentration over potential upon total cell potential is insignificant primarily because of the thin-sized cathode, while the over potential of cathodic activation is smaller by two folds, in comparison to the over potential of ohmic [120–122]. Nonetheless, Hussain et al. [120] asserted the significance of cathode and electrolyte thickness, inclusive of those at the backing and reaction regions in SOFC with anode support, as they could affect the loss in the elements.

For instance, over potential of cathode incorporates activation, ohmic and concentration elements [89]. On the other hand, the ion transport resistance in the electrolyte layer generates electrolyte over potential [120]. Meanwhile, as for SOFC with electrolyte support, electrolyte in the form of thin film is required to decrease ohmic loss for huge ohmic polarization appears to be a rather big threat [123–125].

Furthermore, the illustration in Figure 13 shows that certain porosity and particle size do affect the over potential in electrode minimally based on its thickness [126]. Meanwhile, Figure 13a portrays improvement in cathodic over potential for thickness exceeding minimal setting, which is rather little. This improvement derives from the gas transport resistance mainly because one electrode is thicker than the other that is optimal. Hence, as demonstrated in Figure 13b, as porosity decreases, the curves of the slope are enhanced.

According to Figure 13, when the porosity reduces, the minimum value of the cathodic over potential, decreases. Alternatively, for electrodes thicker than the optimal value, the reduction in porosity may cause a sharp rise in the over potential (Figure 13b). These issues clarify how the optimal thickness decreases in general as the porosity and the particle size reduce (highlighted by the dotted lines in Figure 13).

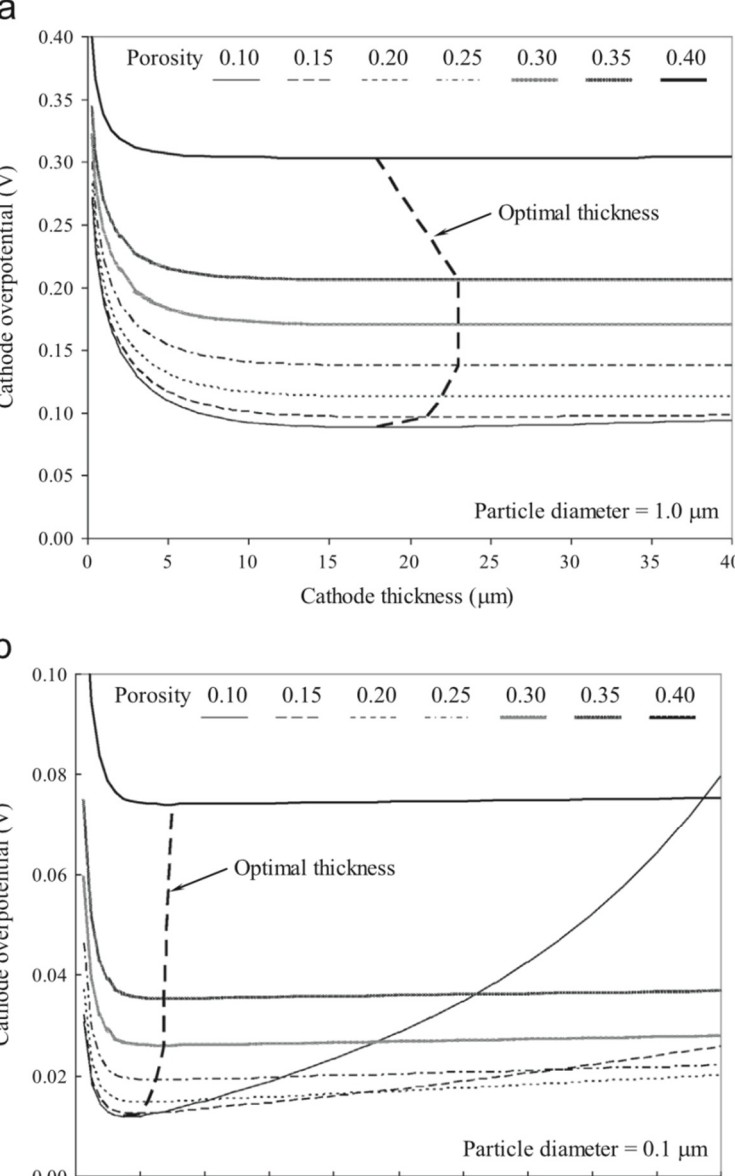

**Figure 13.** Cathodic overpotential as a function of thickness for different porosities: (**a**) particle size equal to 1.0 μm, (**b**) particle size equal to 0.1 μm. In both the simulations, cathodes with equal volume fractions of LSM and YSZ were considered, while the temperature was set to 800 °C (figure reprinted from Bertei et al. [126] Copyright (2013), with permission from Elsevier).

Although considerable progress has been made on the development of simulation and modeling of micro-structure materials and components in recent years, further research is still needed to design and fabricate better fuel cell electrodes and engineer optimum electrode microstructure.

## 4.4. Effect of Fuel Cell Temperature on the Cell Performance

Studying the effect of temperature on fuel cell performance is a key parameter in order to minimize the overall losses, alleviate the thermal stress and improve the SOFC efficiency. Most SOFCs are designed to function at high temperatures, thereby generating higher cell efficiencies in comparison to other kinds of fuel cells [65,127]. The cell current density can be raised by increasing the operating temperature; however, a too high temperature or temperature gradient decreases the cell life time. Therefore, the major challenge is to

optimize the cell durability and to ensure the performance stability of the fuel cell during its operation.

Considering the complexity of the actual energy transport process in SOFCs, simulation is usually a desirable tool to analyze the heat transfer phenomena inside an SOFC and predict its performance. Knowing the temperature distribution within an SOFC is important as it plays a significant role in its operation. Simulation studies focused on thermal distribution and thermal stress were conducted by many investigators to increase the thermal durability and extend the fuel cell life time as well as to determine an optimized operating condition [71,76–80]. Numerical CFD models gave the temperature distribution over the surface of individual cells [89,128], or along a single air channel [103,129]. Since SOFCs operate at high temperatures and the cell scale is tiny, experimental studies on the thermal stresses is difficult. Most advanced research investigations on thermal stresses of SOFCs were performed by establishing computer models and numerical simulations. The issue of designing a reliable SOFC due to high temperature operation was previously detailed in the literature [2,34,35].

There are different methods in the literature to decrease the temperature gradient in fuel cells. Analysis shows that increasing the velocity of the hot air stream, and lowering the Peclet number (by increasing the effective thermal diffusivity of the cell) leads to an optimal design, which minimizes heating time under the constraint of maximum allowable temperature gradients [130]. Significant temperature gradients exist both along the length of the cell and within its cross-section, specifically at the inlet. The high temperatures in SOFCs, further topped with hydrocarbon fuel, generates double reactions, which are: endothermic methane steam reforming reaction and exothermic electrochemical reaction, which happen to occur at various spots in the structure of the PEN [131]. Such huge temperatures, nevertheless, could be brought down with certain percentage recycle of anodic exhaust gases, thus altering the feed composition [127]. Furthermore, the high $H_2$ concentration escalates the reaction of electrochemical, hence leading to a drop in the temperature at the cell inlet [62]. In fact, the high $H_2$, $H_2O$ and CO concentrations could lessen deposition of carbon [127]. Moreover, the high temperatures are usually found in the electrolyte, which is away from channels for flow and unaffected by configuration of SOFC, regardless of electrolyte or anode support [85]. Furthermore, based on estimation of models, temperature for SOFC with electrolyte support is sensitive to over potential heat source, especially at when the gaseous temperature is lower. Nevertheless, an increment in gaseous temperature generates a lower temperature for SOFC with anode support when compared to that with electrolyte support, although functional with similar setting [95].

However, with the decrease in flow channel height, the temperature reduces as well, thus enhancing efficiency of cell mainly because of increased heat/mass transfer coefficient between flow stream and wall of channel, as well as shortened current path [132]. Nevertheless, the bigger cell for fuel that suggests gradual rise in temperature poses risk in relation to thermal stress, which is severe, although it ascertains cell elements' structural integrity. With that, numerous studies have attempted in decreasing the SOFC functional temperature to below 650 °C, especially to avoid degradation of materials, to enhance the lifetime of the layers, as well as to reduce the costs incurred by using common materials that are metallic [17,133–137]. Besides, in order to hinder corrosion that might be severe, especially at higher temperatures, the functional temperature has to decrease, mainly due to the following:

1. Increment in efficiency of thermodynamic conversion can be noted as the temperate for reformed gas (blend of H and CO) is decreased [83].
2. The process of sealing gets intricate as the temperature is lowered [15].
3. Loss of radiation heat is insignificant for a minute system when temperate is decreased. Therefore, management of heat is eased [20].

In fact, another method that could enhance the performance of fuel cell at decreased functional temperature is to lessen the electrolyte thickness [138]. Besides, a fuel cell that

is small could lessen loss of concentration and enhanced performance exerted by the fuel cell [139].

In addition, the variability of SOFC performance depends on an improbable mechanical aspect: the ceramic thin layers are fragile even at moderate stress, which occurs due to several reasons: issues related to fabrication issues or stresses of residual, the variance in TEC (thermal expansion coefficients) layered cell layers, spatial or temporal temperature gradients, oxygen activity gradients, as well as loading of external mechanical. Furthermore, these stresses are affected by the properties of the material, setting of the performance conditions, as well as geometrical design [140]. These residual stresses emerge due to a discrepancy between thermal expansion coefficients and Young's modulus at the layers that adjoin. In fact, such stresses could lead to layers' delamination or cracks in micro-size at the weak layers.

The thermal stresses in the electrolyte are bigger than those of other components for constant temperature distributions. As the temperature rises, the maximum principal stresses in the electrolyte are also enhanced. This occurs as a consequence of the TEC [141].

Based on theory, voltage with reduced functionality offers increased current density and improved density of power, thus hiking both the field and gradient of the temperature. Besides, Table 4 [142] portrays the improvement of thermal stress with decreasing functional voltage meant for anodic porosities at $\varepsilon = 0.2$ and $0.3$, respectively (see also Figure 14). Meanwhile, Table 4 presents higher porosity at $\varepsilon = 0.5$, but achieved thermal stress at the maximum level with intermediate voltage at 0.8 V (Figure 14). Other than that, porosity at $\varepsilon = 0.5$ demonstrated an increment in temperature gradient with 0.8 V voltage, hence generating more thermal stress for PEN. Additionally, the literature [143] dictates that reduced elastic modulus for NiO-YSZ anode, in relation to its Ni-YSZ counterpart, has been due to improved anodic porosity in a direct manner. As such, the decrease of stress due to the hike in anodic porosity (as illustrated in Figure 14) could be a result from the low numbers derived from elastic modulus, shear modulus and Poisson's ratio.

**Table 4.** Maximum principal stresses and temperatures at different cell voltages and porosities [129].

| Prosity | Voltage (V) | PEN (MPa) | Gas Distributors (MPa) | PEN $T_{max}$ (°C) | PEN $\Delta\,T_{max}$ (°C) |
|---|---|---|---|---|---|
| 0.2 | 0.7 | 20.7 | 42.9 | 821.2 | 12.9 |
| 0.2 | 0.8 | 19.6 | 40.5 | 809.9 | 12.8 |
| 0.2 | 0.9 | 18.1 | 37.3 | 808.3 | 12.5 |
| 0.3 | 0.7 | 18.7 | 42.9 | 823.0 | 12.8 |
| 0.3 | 0.8 | 18.4 | 42.1 | 811.3 | 12.6 |
| 0.3 | 0.9 | 16.1 | 36.8 | 806.8 | 11.9 |
| 0.5 | 0.7 | 15.1 | 42.7 | 824.9 | 12 |
| 0.5 | 0.8 | 15.3 | 43.5 | 812.7 | 13 |
| 0.5 | 0.9 | 13.1 | 36.2 | 805.9 | 11.5 |

A critical evaluation of the available bibliography displays that thorough study of the simulation of thermal stresses and the destructive effects of cyclic stresses on the SOFC is still in its early stage of research [144,145]. In most of such reports, the mechanical and thermal physiognomies of the SOFC are taken as constants. According to Nakajo and co-workers [146], however, these characteristics depend mainly on the temperature and the porosity. Furthermore, the temperature changes on the fuel cell surface, so the physiognomies of the materials cannot be considered constant.

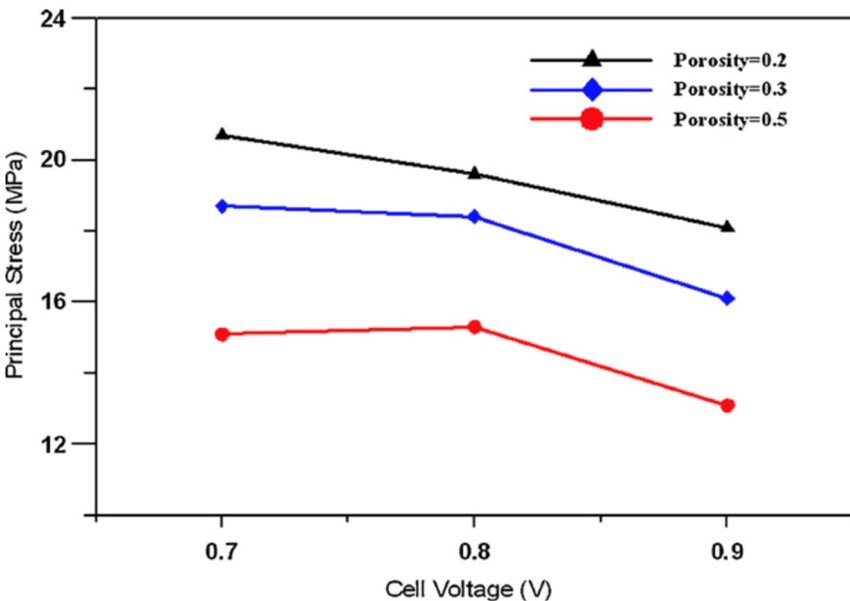

**Figure 14.** Variation of maximum principal stress in PEN with cell voltage for anode porosities of 0.2, 0.3 and 0.5, respectively (figure reprinted from Chiang et al. [142] Copyright (2010), with permission from Elsevier) [143].

## 5. Optimization of SOFC Operational Parameters

Few papers have studied the models of SOFC for the purposes of performance optimization [23,24,90,147], including the steady- and dynamic-states. In order to truly understand the online control strategy for SOFC systems, the optimization of fuel cell operating parameters plays a critical role in improving its performance.

There are models and optimization methods developed by researchers in order to achieve the performance prediction and parameters optimization. The objective of this section is to summarize the present status of such optimization efforts so that unresolved problems can be identified.

Based on the literature, SOFC optimization is classified into: (1) micro-structural, (2) single cell and (3) SOFC hybrid system optimization. In micro-structural optimization, grain size, mean pore diameter, thickness of the cathode/electrolyte/anode functional layer and several other factors are typically considered for optimization [148,149]. For single SOFC optimization strategies, the fuel cell geometry, the design parameters, the cell configuration and the operating conditions were considered [104]. In the single SOFC optimization study performed by Ji et al. [104], a co-flow planar SOFC was studied to figure out the optimum height of the cell. For SOFC hybrid system optimization, different configurations, along with combinations with different equipment and optimum operating conditions were explored [150–152]. A topology of an optimization method was performed to identify characteristics of optimal microstructures of a nano-composite SOFC cathode produced by infiltration of a mixed ionic electron conductor phase into a porous ionic conducting scaffold [153].

In order to optimize fuel cell systems to work in an efficient manner with the least cost and highest power generation, different optimization methods like genetic algorithm [154], clustering Pareto evolutionary algorithm [155], design of experiments [156] and also different methods of multi-objective optimization (MOO) have been investigated. MOO techniques provide the necessary information for a detailed examination of the design and operational trade-offs between conflicting objectives. However, only a few reports address the issues of MOO of SOFC stacks.

A multi-objective evolutionary algorithm (Pareto envelope-based selection algorithm) was used for a combination of photovoltaic panels, batteries and an SOFC using different fuels fed to the latter to determine enough auxiliary power for this hybrid system economi-

cally and ecologically [157]. The most desirable hybrid system was the one that had the least cost, the least emission and the most reliability. However, these objectives were in conflict with each other. It was reported that the most appropriate fuel for the SOFC was the hybrid system involving natural gas due to low annualized cost as well as low $CO_2$ emissions.

Quddus et al. [158] presented multi-objective trade-offs in SOFCs for oxidative coupling of methane. Palazzi et al. [159] performed MOO on an SOFC system. The proposed objectives were efficiency and capital cost per unit power ($/kW). Chakraborty [154] used a genetic programming approach in an SOFC system. Yang et al. [160] presented an improved genetic algorithm to optimize the parameters in the simple SOFC electrochemical model. The results of the optimization parameters showed that the improved genetic algorithm had more precision and stability compared to its simple genetic counterpart.

Nonetheless, in order to enhance one's comprehension pertaining to the performance exerted by the SOFC system, the influence of several significant parameters, as well as the correlations between these parameters in association to the performance of the general system, were looked into by employing empirical optimization [156]. Moreover, the enhanced iterative particle swarm optimization (PSO) algorithm was calculated to determine the SOFC-micro gas turbine hybrid system, especially to seek optimization among the functional parameters with varied loads [161]. In fact, this particular approach is amalgamated with the technique of iteration and the PSO algorithm, thus generating discrete PSO in an iterative manner right until convergence takes place from the control profile to one that is optimal. Other than that, the findings obtained from the MATLAB simulation display that the hybrid model of SOFC/MGT, together with parameters that were optimized, could effectively and efficiently determine power output. In short, the enhanced iterative PSO algorithm could prove to be beneficial when applied for analysis of systems, optimization of designs, as well as real-time control for a hybrid system, in particular, the SOFC/MGT system [161].

A useful method involving thermodynamic optimization has been performed by Bejan [162]. The purpose of thermodynamic optimization is to minimize losses, thereby maximizing power output and efficiency. The two thermodynamic optimizations developed by Bejan are entropy generation minimization and constructal theory. Sciacovelli [163] modeled a single channel of a monolithic cell using CFD (Fluent$^{TM}$) and simulated a stack that also considered a thermal model. The total entropy decreased by 50% using the EGM method which resulted in a power density increase of 10%.

Improvement of fuel cell design results in enhancement of power density. Reducing the anodic thickness first results in an increase of power density due to a decrease of ohmic and gas-phase transport losses. By reducing anodic thickness further, the layer becomes too thin to support the electrochemical conversion. The thickness of the cathode has almost no influence on the system's performance because the cathodic contribution to the power loss of the system is not important. In this regard, Vogler et al. [164] identified an optimum anodic thickness to improve the fuel cell power.

Maximization of power at minimum emission of greenhouse gases is an issue. Therefore, a multi-objective optimization study using non-dominated sorting genetic algorithm is performed to determine the optimal operating conditions and design configurations of the SOFC unit. In this regard, the SOFC power plant is optimized in terms of maximizing the $C_2$ production and minimizing the generation of undesired products such as $CO_x$ or $FCO_x$ [158]. A summary of the objectives for fuel cell optimization is listed in Table 5.

Large-scale power generation benefits from the high efficiency of gas-steam combined cycles. In the lower power range, fuel cells especially SOFCs operating at high temperatures are good candidates to combine with gas turbines to achieve efficiencies exceeding 60%. A configuration of hybrid system technology has been identified using a thermo-economic multi-objective optimization approach [165]. The system is integrated using pinch based methods. Then, a thermo-economic approach is used to compute the integrated system performances, following which, two objectives are optimized: (1) minimization of the

specific cost, and (2) maximization of the efficiency. The optimization results in a design with costs ranging from $2400/kW for a 44% efficient system to $6700/kW for a 70% efficient operation [165]. The PSO method is used for a hybrid power system to determine the optimum power sharing means to enhance the total efficiency of the system [166].

**Table 5.** A summary of objectives for fuel cell optimization.

| Objective | Constraint | Optimization Method | Refs. |
|---|---|---|---|
| Maximize performance of power production from waste biomass | Temperature | Clustering Pareto evolutionary algorithm | [155] |
| Maximize the system efficiency | Operating pressure | Design of experiments | [156] |
| Maximization of system efficiency and the minimization of specific investment cost | Fuel processing temperature Steam to carbon ratio Fuel utilization Oxygen to carbon ratio Air excess ratio | Based on the use of an evolutionary Algorithm: MOO (Multi-Objective Optimizer) | [159] |
| Parameter optimization: Maximize the output voltage and current density to find cathodic shapes that minimize resistance at the base of the cell | Pressure, temperature | Improved genetic algorithm Finite element modeling for topology optimization | [160] |
| Reducing the annual total cost of power, Reducing the heating and cooling generation and the annual $CO_2$ emissions rate | The amount of fuel provided to the SOFC; The operating temperature of the SOFC; the operating pressure of the SOFC | Evolutionary Multi-objective optimizer | [161] |

## 6. Summary and Recommendation

In recent years, considerable progress has been made in SOFC simulation to improve the design and performance of this technology with the aim of reducing its cost. In this work, different designs and configurations of SOFCs (an essential step in simulation to represent the real system as closely as possible) were reviewed. Despite that, such SOFC simulations require much attention to enhance the performance of the system.

The main conclusions of this review are as follows:

- SOFC materials and micro-structures were reviewed due to their significant impact on the simulation reliability as well as system optimization. A literature review has also been performed on simulation studies focused on the effects of microstructure and materials on fuel cell performance. However, most of the studies use only common SOFC materials in their modelling. More simulation studies are required to compare alternative materials in overall modelling to assess their effect on the efficiency and degradation of the cell. Electrochemical modelling that studies the cell voltage behaviour as a function of the microstructure, geometry and material properties is still under development.

- In the literature, different techniques of SOFC simulations are available. In cell and stack level, 0-D, 1-D, 2-D and 3-D techniques may be chosen according to the purpose of the model. Transient modelling is used if any of the heat-up, start-up, shut-down and load changes or a combination of them requires simulation.

- A critical analysis of available literature proved that detailed research on the simulation of thermal stress and damaging impact upon the SOFC is still in its early stage of development. In most of the presented works, the mechanical and thermal characteristics of the SOFC are represented comprehensively [140,144,146]. There is a lack of simulation reports, however, on the radiative heat transfer between the materials used in SOFCs at the high temperatures. In addition, the literature lacks sufficient

information on the energy and exergy analyses of SOFC systems for performance evaluation.
- In order to understand online control and optimization strategies accurately, an effective simulation of the fuel cell system plays a critical role. Therefore, sensitivity analysis of fuel cell parameters using simulation models was reviewed herein. In addition, micro-machined SOFCs are used for portable devices (many electronics and wireless). Hence, optimization and reliability of such devices require further research.
- Combination of an MPS, a deep neural network and a genetic algorithm provides a promising solution for model-based control systems for precisely and quickly analyzing the performances of SOFCs and other non-linear systems. The deep learning algorithm needs to be trained based on experimental data for a wider application under dynamic operating conditions.

**Author Contributions:** Conceptualization, M.T. and A.S.; writing—original draft preparation, M.T.; writing—review and editing, A.S. All authors have read and agreed to the published version of the manuscript.

**Funding:** This research received no external funding.

**Institutional Review Board Statement:** Not applicable.

**Informed Consent Statement:** Not applicable.

**Data Availability Statement:** Not applicable.

**Conflicts of Interest:** The authors declare no conflict of interest.

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
