# Peer review of "Simulation and Sensitivity Analysis for Various Geometries and Optimization of Solid Oxide Fuel Cells: A Review"

_2673-4117, doi:10.3390/eng2030025_

Round 1

Reviewer 1 Report

Comments and Suggestions for Authors

Dear Authors,

The Title: Simulation and sensitivity analysis for various

geometries and optimization of solid oxide fuel cells: a

review

I have to read your manuscript with great attention and interest. The material is consistent, comprehensive and complete.

Recommendation that all diagrams and drawings should be prepared according to the same graphic design throughout the manuscript, after placing the title, add the information that it has been modified in the sub-presentation and add that the Author has consented to the publication of the data.

The submission falls within the scope of the journal and is sufficiently original, and I have a remark, so I recommended the publication after MAJOR REVISIONS.

Fig. 2. Add the information about the data e.g. Dec. 2020

Fig. 3. to whom these photos belong, do not need the Authors to consent to their publication

Fig 3 c. Add the scale

Table 1. Add the References in the Table

Fig. 6. Please prepare your own diagram processing, please correct the H2 / H2O data and the direction of input or output of media, check if the Rolls-Royce concern use N2O2

 Fig. 7. Improve the quality of the scheme

Fig. 9. You can add a match or a ruler to show the size of the micro-tubular fuel cell

Fig. 13 There is no call in the text

Summary: What conclusions does the knowledge contained in the manuscript bring for future research of the Authors? Prepare the Conclusion in points.

Author Response

Responses to the reviewers’ comments on the manuscript

The authors would like to thank the respected reviewers for the careful review of our manuscript and for providing us with their valuable comments and suggestions to improve the quality of the manuscript. The following responses have been prepared to address all of the reviewers’ comments in a point-by-point fashion in blue and the manuscript was revised accordingly. Changes in the manuscript are marked in red.

Reviewers' comments:

Reviewer #1

I have to read your manuscript with great attention and interest. The material is consistent, comprehensive and complete.

Recommendation that all diagrams and drawings should be prepared according to the same graphic design throughout the manuscript, after placing the title, add the information that it has been modified in the sub-presentation and add that the Author has consented to the publication of the data.

The submission falls within the scope of the journal and is sufficiently original, and I have a remark, so I recommended the publication after MAJOR REVISIONS.

The authors would like to thank the respected reviewer for the careful review of our manuscript and for providing us with the valuable feedback.

Comment 1:

Fig. 2. Add the information about the data e.g. Dec. 2020

Response 1: 

The figure caption is revised accordingly as follows:

Fig. 2: Number of citations of articles per year (untill December 2020) related to “SOLID OXIDE FUEL CELL OPTIMIZATION”, showing the increasing research interest in this topic. Data from Scopus.

Comment 2:

Fig. 3. to whom these photos belong, do not need the Authors to consent to their publication

Response 2: 

This figure is a new and combined figure and does not belong to any authors. Therefore, we haven’t added any reference for that.

Comment 3:

Fig 3 c. Add the scale

Response 3: 

Unfortunately, we are not able to provide an accurate scale for that.

Comment 4:

Table 1. Add the References in the Table

Response 4: 

We have summarized the characteristics of different self-supported SOFCs and therefore, we have not added any reference in this table.

Comment 5:

Fig. 6. Please prepare your own diagram processing, please correct the H2 / H2O data and the direction of input or output of media, check if the Rolls-Royce concern use N2O2

Response 5: 

The figure is revised based on the reviewer’s comment.

z

Comment 6:

 Fig. 7. Improve the quality of the scheme

Response 6: 

The quality of figure 7 is improved as follows:

Fig. 7: Schematic of BSC with micro-channels (figure reprinted from Gardner et al. [47]  Copyright (2000), with permission from Elsevier).

Comment 7:

Fig. 9. You can add a match or a ruler to show the size of the micro-tubular fuel cell

Response 7: 

It is not suitable to show the size of micro-tubular fuel cell using match or a ruler.

Comment 8:

Fig. 13 There is no call in the text

Response 8: 

We did not understand the reviewer’s comment and haven’t seen any issue with this figure.

Comment 9:

Summary: What conclusions does the knowledge contained in the manuscript bring for future research of the Authors? Prepare the Conclusion in points.

Response 9: 

The summary and recommendation section is revised based on reviewer’s comment as follows:

In recent years, considerable progress has been made in SOFC simulation to improve the design and performance of this technology with the aim of reducing its cost. In this work, different designs and configurations of SOFCs (an essential step in simulation to represent the real system as closely as possible) were reviewed. Despite that, such SOFC simulations require much attention to enhance the performance of the system.

The main conclusions of this review are as follows:

-SOFC materials and micro structures were reviewed due to their significant impact on the simulation reliability as well as system optimization. A literature review has also been performed on simulation studies focused on the effects of microstructure and materials on fuel cell performance. However, most of the studies use only common SOFC materials in their modelling. More simulation studies are required to compare alternative materials in overall modelling to assess their effect on the efficiency and degradation of the cell. Electrochemical modelling that studies the cell voltage behaviour as a function of the microstructure, geometry and material properties is still under development.

-In the literature, different techniques of SOFC simulations are available. In cell and stack level, 0-D, 1-D, 2-D, and 3-D techniques may be chosen according to the purpose of the model. Transient modelling is used if any of the heat-up, start-up, shut-down, and load changes or a combination of them requires simulation.

-A critical analysis of available literature proved that a detailed research on the simulation of thermal stress and damaging impact upon the SOFC is still in its early stage of development. In most of the presented works, the mechanical and thermal characteristics of the SOFC are represented comprehensively [140,144,146]. There is a lack of simulation reports, however, of the radiative heat transfer between the materials used in SOFCs at the high temperatures. In addition, the literature lacks sufficient information on the energy and exergy analyses of SOFC systems for performance evaluation.

-In order to understand online control and optimization strategies accurately, an effective simulation of the fuel cell system plays a critical role. Therefore, sensitivity analysis of fuel cell parameters using simulation models was reviewed herein. In addition, micro-machined SOFCs are used for portable devices (many electronics and wireless). Hence, optimization and reliability of such devices require further research.

- Combination of an MPS, a deep neural network, and a genetic algorithm provides a promising solution for model-based control systems for precisely and quickly analysing the performances of SOFCs and other non-linear systems. Deep learning algorithm needs to be trained based on experimental data on for a wider application under dynamic operating conditions.

Reviewer 2 Report

In general paper is well written and can be considered for publication after revision.

  1. The paper is titled "Simulation and sensitivity analysis for various geometries and optimization of solid oxide fuel cells: a review" however almost half of text is description of various SOFC types. Considering paper title reader can expect to have more detail review focusing on “Simulation and sensitivity analysis” so I recommend to authors to include more information in the sections 3-5. The related problem is that this review does not cite recent works (there is only 1 reference from 2021, nothing from 2020 and only one 2019). Recent work related with “Simulation and sensitivity analysis” should be included. See for example https://doi.org/10.1016/j.energy.2020.117752, doi: 3389/fchem.2020.609338, https://doi.org/10.1016/j.egyai.2020.100003 or https://doi.org/10.1016/j.rser.2010.12.011.
  2. In some table like Table 1 sometimes it is unclear which terms from 2nd column are related with cell configuration types. The distance between cell rows should be added.
  3. Fig 3b. The caption said that it is cathode supported cell but from Fig. we can thin top cathode layer. If it is cathode supported cell why cathode layer is so thin?
  4. Caption Fig. 2.It is unclear what do author mean by “citation”. Is it number of papers which have inside word combination “SOLID OXIDE FUEL CELL OPTIMIZATION”?
  5. Section 5. It is unclear why “a review” is included in section title.
  6. I recommend to revise reference list since some references (6, 39) are citing same paper.

Author Response

Responses to the reviewers’ comments on the manuscript

The authors would like to thank the respected reviewers for the careful review of our manuscript and for providing us with their valuable comments and suggestions to improve the quality of the manuscript. The following responses have been prepared to address all of the reviewers’ comments in a point-by-point fashion in blue and the manuscript was revised accordingly. Changes in the manuscript are marked in red.

Reviewer #2

Comments and Suggestions for Authors

In general paper is well written and can be considered for publication after revision.

Response:

The authors would like to thank the respected reviewer for the careful review of our manuscript and for providing us with the valuable feedback.

Comment 1:

The paper is titled "Simulation and sensitivity analysis for various geometries and optimization of solid oxide fuel cells: a review" however almost half of text is description of various SOFC types. Considering paper title reader can expect to have more detail review focusing on “Simulation and sensitivity analysis” so I recommend to authors to include more information in the sections 3-5. The related problem is that this review does not cite recent works (there is only 1 reference from 2021, nothing from 2020 and only one 2019). Recent work related with “Simulation and sensitivity analysis” should be included. See for example https://doi.org/10.1016/j.energy.2020.117752, doi: 3389/fchem.2020.609338, https://doi.org/10.1016/j.egyai.2020.100003 or https://doi.org/10.1016/j.rser.2010.12.011.

Response 1:

Based on reviewer’s comment, the following statements are added to sections 3 and 4 of the revised manuscript and highlighted in the red color.

Section 3:

Eichhorn Colombo et al. [81] employed a detailed solid oxide fuel cell (SOFC) model for micro-grid applications to analyze the effect of failure and degradation on system performance. They presented the design and operational constraints on a component- and system level. A degrees of freedom analysis identifies controlled and manipulated system variables which are important for control. They used experimental data to model complex degradation phenomena of the SOFC unit. They assumed that the SOFC unit is consisting of multiple stacks. They studied the failure scenario of the loss of one individual SOFC stack, e.g. due to breakage of sealing or a series of fuel cells. Their simulation results reveal that degradation leads to significant drifts from the design operating point. In addition, failure of individual stacks may bring the still operating power generation unit into a regime where further failures and accelerated degradation is more likely. It is presented that system design, dimensioning, operation and control are strongly linked. Apart from specific quantitative results perhaps the main practical contribution are the collected constraints and the degrees of freedom analysis.

Haoran et al. [82] developed a hybrid model for an on-line analysis of SOFCs at the cell level. Their model combines a multi-physics simulation (MPS) and deep learning, overcoming the complexity of MPS for a model-based control system, and reducing the cost of building a database (compared with the experiments) for the training of a deep neural network. They considered the maximum temperature gradient and heat generation as two target parameters for an efficient operation of SOFCs. Their modelling results reveal that a precise prediction can be achieved from a trained AI algorithm, in which the relative error between the MPS and AI models is less than 1%. Furthermore, an online optimisation is realised using a genetic algorithm, achieving the maximum power density within the limitations of the temperature gradient and operating conditions. Their method can also be used for prediction and optimisation of other non-liner, dynamic systems. Their study shows that the combination of an MPS, a deep neural network, and a genetic algorithm provides a promising solution for model-based control systems for precisely and quickly analysing the performances of SOFCs and other non-linear systems. Their deep learning algorithm is trained based only on simulation data under stationary operating conditions. Further training with more experimental data on the algorithm is needed for a wider application under dynamic operating conditions.

Section 4:

Dang et al. [99] developed and solved a numerical model of gas flow, heat transfer, mass transfer and electrochemical reaction multi-physics field coupling of a planar SOFC. They analyzed distribution of velocity, temperature and concentration inside the SOFC cell. They discussed the influence of cathode inlet flow rate, porosity, rib width and other parameters on the performance of SOFC. Their simulation results showed that within a certain range, increasing the cathode inlet flow rate can significantly increase the average current density of the cell. Increasing the porosity of the electrode can improve the gas diffusion of the porous electrode, thereby increasing the rate of the electrochemical reaction. Increasing the width of the ribs will result in a significant decrease in cell performance. Therefore, the rib width should be reduced as much as possible within the allowable range to optimize the working performance of the cell.

Comment 2:

In some table like Table 1 sometimes it is unclear which terms from 2nd column are related with cell configuration types. The distance between cell rows should be added.

Response 2:

Based on reviewer’s comments, the tables are revised accordingly.

Comment 3:

Fig 3b. The caption said that it is cathode supported cell but from Fig. we can thin top cathode layer. If it is cathode supported cell why cathode layer is so thin?

Response 3:

The cathode is the bottom layer.

Fig 3b. was revised as follows:

Cathode

Anode

Electrolyte

Comment 4:

Caption Fig. 2.It is unclear what do author mean by “citation”. Is it number of papers which have inside word combination “SOLID OXIDE FUEL CELL OPTIMIZATION”?

Response 4:

The figure caption is revised as follows:

Fig. 2: Number of citations of articles per year (untill December 2020) related to “SOLID OXIDE FUEL CELL OPTIMIZATION”, showing the increasing research interest in this topic. Data from Scopus.

Comment 5:

Section 5. It is unclear why “a review” is included in section title.

Response 5:

Section 5 title is revised as follows:

  1. Optimization of SOFC operational parameters

Comment 6:

I recommend to revise reference list since some references (6, 39) are citing same paper.

Response 6:

The reference list is revised based on reviewer’s comment.

Round 2

Reviewer 1 Report

The authors improved most of the recommendations. I accept in present form.